# Radon at Kilbourne Hole Maar and Magnetic and Gravimetric Correlations

**DOI:** 10.3390/ijerph20065185

**Published:** 2023-03-15

**Authors:** Michel E. Luna-Lucero, Laszlo Sajo-Bohus, Jorge A. Lopez

**Affiliations:** 1Department of Physics, University of Texas at El Paso, El Paso, TX 79968-0515, USA; meluna4@miners.utep.edu; 2Departamento de Física, Universidad Simón Bolívar, Caracas 89000, Venezuela; lsajo@usb.ve

**Keywords:** underground radon, Kilbourne Hole, Potrillo Volcanic Field, magnetic anomalies, gravimetric anomalies, maar

## Abstract

Soil radon gas concentrations ranging from the detection limit up to 15 kBq/m^3^ were measured for the first time at the Kilbourne Hole maar in two selected regions: the first region was located on the western volcanic field, and the second was located inside the crater, near the southern border. Radioactive anomalies were found in association with the pyroclastic deposit, and the corresponding heat map provided information on the radon diffusion direction by the C_Rn_ gradient. It was observed for the first time that the anomalies found at the southern border are associated with a known geological fault, in opposition to what was found on the western border. The results provided by a radon activity concentration gradient of above (8 kBq/m^3^)/15 m suggest the existence of a fault that has not been detected yet. The observation that high levels near a dormant fault are related to tectonically enhanced radon was confirmed. The activity concentrations of Rn-gas were contrasted to existing gravimetric and magnetic data to provide measuring information on radon emanation, suggesting the existence of a high, naturally occurring radioactivity in the soil in the first place or an increased porosity of the locally defined lithology. The results indicated a higher correlation of 85% with magnetic anomalies. This is in opposition to the gravimetric data, which was only 30%. This study is a contribution to the characterization maar of volcanic geology by the soil radon activity index, which was designated as “low” in this case.

## 1. Introduction

Geophysical studies may take advantage of naturally occurring radioactive matter (NORM) such as radon (Rn) isotopes and their progeny. Radioactive gas can be measured at or near the Earth’s surface soil and air interface using, e.g., a suitable monitor. Variations in radon emanation from the soil yield information about the Earth’s subsurface crust that can be used advantageously in studies conducted at volcanic areas and geological fault lines around the world. Miklavčič et al. [1] determined radon anomalies at the sites Osijek and Kašina, Croatia, and proposed an empirical equation to connect radon concentration C_Rn_ values to variations in barometric pressure, rainfall, and temperature.

A relation between radon gas and fault lines was observed by, e.g., Baykara et al. [2], who determined the profile of soil radon gas in the North and East Anatolian active fault system in Turkey, and more recently by [3]. Data analysis shows that soil radon gas activity concentration (C_Rn_) values are markedly high near or above a fault line in comparison to the adjacent soil. A decreasing Rn value that moves away from the preselected lines could be employed to identify a radioactive spot. These values correlate well with underlying bedrock morphology and provide a method to localize tectonic discontinuities [4].

A previous study on C_Rn_ around a fault at the East Franklin Mountains (El Paso, Texas) indicated strong correlation between the in-soil concentration of radon gas and, as mentioned, the location of a geological fault close to an inactive volcano [5,6].

The correlation between the C_Rn_ in top soil (radioactive anomaly) and Earth’s magnetic field intensity was reported by Girault et al. [7]. It is worth noting that this is not a direct correlation, as Rn is inert and is not a magnetic substance. It is, however, a correlation through the soil matrix magnetism (minerals, organic matter, air, and water). Rusov et al. [8] determined a relationship between the radon concentration and temporary changes in the Earth’s magnetic field resulting from plate tectonic movements or related activities. Temporal variations in radon concentration correlated to groundwater flow were determined by Perrier et al. [9], a phenomenon that could explain the formation of radioactive anomalies or variations of soil gas radon concentrations over different geological formations of several kBq/m^3^. On average, the C_Rn_ at a relaxation depth of 35 cm is approximately 5.1 kBq/m^3^ [6]; when the measured value is above several kBq/m^3^, a so-called radioactive “hot spot” is used to identify the most suitable place for further measurements.

Geomagnetic, decline-anomalous phenomena could be a consequence of past volcanic activity; similarly, changes in gravity and radioactive concentration could be related to a dormant volcano. Measurements of C_Rn_ values provide basic information to further inquire upon volcanic region characteristics. Through isocurve contour gradients, the resulting values provide information to demonstrate the radon transport direction, which can be employed to establish, e.g., a possible correlation to the crust magnetism and gravimetric data.

This study will introduce several novelties. (i) Radon activity concentration gradients will be used to identify a geological fault inside of a maar crater. (ii) Radon activity will be correlated to geomagnetic anisotropy and to Bouguer gravity anomaly. To our knowledge, this is the first time this has been achieved in the crater of a maar.

### 1.1. Radon in Volcanogenic Deposits

All natural elements that are heavier that lead are radioactive; some of them are related to radon gas. Primordial radioactive families (4n; 4n + 1; 4n + 2 and 4n + 3) correspond to the ^232^Th(^220^Rn), ^237^Np(^220^Rn), ^238^U (^222^Rn) and ^235^U (^219^Rn) decay chain (DC). Radon isotopes often actively employed in geophysical studies include ^222^Rn and its progeny, e.g., ^218^Po; ^214^Po.

The C_Rn_ in a volcanogenic deposit, i.e., in or near volcanic rocks, is determined by measuring DC products. Thomas studied rocks formed by explosive volcanism such as tuff and breccia, observing that they contained natural U as uraninite [10]; Masuda et al. [11] reported back in 1971 that uranium is adsorbed in volcanic ash soils.

These results are indications that due to the presence of uranium and its progeny, radioactive anomalies have a high occurrence near the soil surface (or surficial radioactive gas). Surficial C_Rn_ levels could change by three main factors: the concentration of naturally occurring radioactive matter (NORM), radioactive matter convective transport, and a layer of rock or sediment characterized by known lithological properties, including bedding surfaces (planes).

The world’s average soil NORM concentration of uranium is 30 Bq/kg (or 3 ppm); thorium is four times higher, and radium has the largest variation, contributing to radon generation at a rate of 10^−3^ cm^3^/d per g of ^226^Ra (4n + 2 DC). However, due to tectonic dynamical behavior, such as the behaviors manifested by volcanoes and other geological events, C_Rn_ levels may differ by orders of magnitude. This relatively high value found in the geological strata is attributable to the presence of uranium (with a concentration of ~475 ppm) and to the low solubility of the metal in local groundwater.

The other transport phenomenon that may induce radioactive “hot spots” with a relatively high C_Rn_ value is related to the surficial accumulation of ^226+228^Ra, as it is the heaviest of the alkaline earth metals group. This radioactive element with no stable atom forms carbonate compounds and is susceptible to migrate, i.e., to be transported by underground fluid motion.

Therefore, underground radium radionuclides show a large concentration variability of up to 3000 Bq/kg. Consequently, as an anomaly accumulation, its occurrence is an expected event at Kilbourne Hole (as previously mentioned by an explosive eruption); these expectations are supported by the studies of Cothern and Smith [12].

### 1.2. Study Site and Objective

The site, which was specifically selected for its geological characteristics, is Kilbourne Hole. Kilbourne Hole is a broad volcanic crater formed by an explosive eruption. The maar volcano, which emitted lava to form a shallow rim and pyroclastic fragments, is a phreatomagmatic, bowl-shaped depression. The sunken landform originated from the interaction of rising magma with an aquifer or surface water [13]. Its extension is approximately 1.7 miles long by one mile and a quarter wide, with a floor between 250 to 300 feet below the general level of the surrounding setting. The crater has a border rim that rises up to 150 feet [14]. The La Mesa is found at the center, 450 ft. above the Playa floor [15]. It is integrated into the Potrillo Volcanic Field that stretches from Dona Ana County, NM, USA, into the northern region of Chihuahua, Mexico [16].

We previously mentioned the usefulness of the knowledge of radon or radioactivity concentration variations for structural geology characterization. It is well documented that for gases crossing uranium-rich underground lithologies, fractures or discontinuities often constitute a favorable exit route to the Earth’s surface. Here, the convenience to employ soil radon gas measurements in the case of Kilbourne Hole is justified since its formation is the consequence of mass displacements, with a favorable location on the north-trending fault of the Fitzgerald–Robledo fault system. Expected Rn anomalies at the crater will provide new information to determine local characteristics. For instance, the degree of correlation between the concentration of surficial soil radon gas, and local geophysical characteristics such as the magnetic field and gravity anomaly.

In summary, the question to answer is, what is the radon production in the Kilbourne Hole maar? In addition, given that the magnetic anomaly and geology anisotropy of the maar are known, do they have any effects on the radon concentration gradients? In this case, the research gap is the missing information on radon concentrations in a maar, as well as radon gradient changes within existing anomalies in a given magnetic field and defined geological stratum. Furthermore, this study confirms the possibility of identifying geological faults through the measurement of Rn.

## 2. Materials and Method

The concentration of the underground radon gas, C_Rn_, was measured using the Markus-10, a portable monitor manufactured by Gammadata in Sweden. The Markus 10 instrument works by drawing air into a detector chamber, which captures the radon decay products, measures the energy and number of the particles, and determines the concentration of radon in the air sample. Its lower limit of detection (LLD) is as low as a few becquerels per cubic meter of air.

Figure 1A shows the decay chain of ^222^Rn into ^218^Po, which occurs with a half-life of 3.8 d, followed by the alpha decay of ^218^Po into lead in only three minutes; the alphas coming from the ^218^Po decay are the ones detected to determine the Rn concentration.

Figure 1B shows the Gammadata Markus 10 instrument used to measure the radon concentration in soil. The device consists of:A stainless steel probe that is hammered down into the soil at a depth of 70 cm, a length selected because allows for 80% of the C_Rn_, to reach the surface–air border by diffusion [18];An air pump to which the probe is coupled through an airtight connector to allow for the flow of only the soil gas. The pump capacity of the Markus 10 instrument is 1.8 L/min;A counting chamber in which soil gas is accumulated for 30 s (effective pumping time), followed by a 10 min. measuring time to allow for ^218+214^Po accumulation;A large-area silicon surface barrier detector (SSB) that is coupled to a specific electronic circuitry to count the rate of alpha particles. The instrument displays the average C_Rn_ value in kBq/m^3^.

Figure 2 shows the location of the selected inspection sites. They can be grouped into three clusters of radon stations, referred to here as C_Rn_ TPs. Group (a) of 31 TPs, positioned at the crater surface every 250 m, were established in the west–east direction and in 500 m intervals along the north–south trajectory. Group (b) of 23 TPs were established on the borderline limiting the crater surface, and group (c) of 14 TPs was established along a straight line.

Group (c) was selected to investigate the Rn emission across a dormant fault (see the sequence of red symbols at the bottom of Figure 2). The longitude and latitude coordinates were determined using a Garmin^®^ GPS that provides an accurate position within 5 to 10 m through the processing of multiple signals (GPS, GLONASS, etc.). Finally, the measurements were taken on 10, 14, and 16 October, and 31 December of 2020 and on 3 and 5 February of 2021.

## 3. Results

The C_Rn_ was recorded for 53 TPs (radon stations), providing the heat map given in Figure 3. Measurements at 14 TPs were carried also out across the southern inactive fault (see South Fault map provided), and the results are reported in the heat map. In the heat map, we indicate C_Rn_ values with a continuous line to evidence large gradients.

As mentioned, the Markus instrument shows the C_Rn_ value. The value is displayed by a LED readout, and the associated error is of 1.7%. Measured values are rounded off to tenths of kBq/m^3^. Consequently, any displayed value could be ±0.5 kBq/m^3^ of the value expected; for instance, the C_Rn_ values falling between 13 and 14 kBq/m^3^ could indeed correspond to any value in the interval.

The radon concentration highest values are located inside of the crater, with a maximum radon concentration recorded at 15 kBq/m^3^. This value exceeds the average radon level at surrounded the crater by approximately 10 kBq/m^3^. The difference of these values may indicate the influence of the geologic bodies and/or geological structure.

The radon stations were placed at a few meters from one to another to cover a line that crossed the south fault (see Figure 4). The profiles of radon concentration in the soil are shown in Figure 4 versus the position of stations along the line crossing the fault, where the increase in the radon level with the location of the fault can be observed. The profile of Figure 4 clearly demonstrates the relationship between the radon levels and the inactive faults.

## 4. Discussion

This section explores three results of our study, with emphasis on possible implications due to the soil gas radon concentration in association with the Earth’s magnetic field and the available gravimetric data for the selected dormant volcanic site.

### 4.1. Soil Radon Gas Surficial Survey

The heat map (Figure 3) clearly shows a radon concentration inside of the crater that is much larger than the surrounding areas. This suggests the possibility that the material inside the crater has a different composition and structure (e.g., porosity) than the material of the neighboring zone, calling for further analysis.

Likewise, the study of the south fault (c.f. Figure 4) clearly shows an increase in the radon concentration near the fault. Indeed, Figure 4 shows that the radon levels rise markedly by a factor of 10 at some TPs along the existing fault location. This confirms the observations of Rodríguez and López [5,6], which indicated that the concentration of underground radon gas increases near geological faults. Furthermore, the high levels of Rn found at the northeast rim of the crater could indicate the existence of an undetected fault.

As it is known that the flow of radon concentration is influenced by the water content in the soil, it is convenient to mention the water conditions present at Kilbourne Hole during the measurements. Since the radon diffusion coefficient decreases as the water content increases [19,20,21], the presence of water in the soil would indicate that our measurements correspond to a lower bound. However, the dates of the measurements, October 2020 to February 2021, correspond to the low-rain season, which has a peak between June and September, and no rain was observed in the area on those dates. Likewise, when looking at the USGS topographic map of Kilbourne Hole [22], a “dry lake” depression at 3924 ft (1196 m) of altitude lies near the center of the crater, while the radon maxima extends from the south of the lake (at 3950 ft or 1203 m), across the lake, and into the rim at higher ground (at 4000 ft or 1219 m). Thus, we do not observe any effect on the radon measurements due to the variation in altitude and the possible existence of water contained underground.

As indicated at the beginning of this Section, more analysis is needed to tie the high Rn levels to the materials inside the crater. In the next section, we compare C_Rn_ with the magnetic anomalies and gravimetric data of Kilbourne Hole. Here, we use physical geography to examine the relationships between radon concentration, magnetic anomalies, and gravimetric data.

### 4.2. Soil Radon Gas Association with Gravity Anomalies

A gravimetric survey of Kilbourne Hole provided data on the soil density variations measured at 172 TP. For these purposes, a LaCoste and Romberg gravity meter was employed to measure the acceleration due to gravity. The position of each station, arranged on a grid with 200 m spacing, was obtained using a Topcon GB-1000 real-time kinematic (RTK) differential unit [23]. The resulting map of Kilbourne Hole, provided in Figure 5, delineates three distinct anomalies, i.e., (1) the diatreme and dikes; (2) the Camp Rice formation and older units that predate the eruption; (3) buried basalt lava flows. The value in the total gravity anomaly between (1) and (2) is ~40 mGal, whereas the contrast between (1) and (3) is ~20 mGal. The white line delimited by circles (o) defines the region of the gravity anomaly (see Figure 5a). For comparison, the contour of region A is provided in Figure 5b.

The degree of correspondence of the C_Rn_ anomaly with the gravity anomaly provides a clear view of the overlapping degree for the selected regions indicated by A and B; the analysis indicates that ~30% corresponds to the intersecting area, (A∩B). The value was determined by employing the sum of arbitrarily sized pixels, using 72 pixels for surface area A, 109 for B, and 22 for A intersecting B.

### 4.3. Correlation of Soil Gas Radon Map with Geomagnetic Field

The magnetic map was drawn using 166 measurements, carried out on a grid with 200 m spacing. The employed instrument consisted of a single-sensor-type G858 based on a cesium-vapor magnetometer [23]. The displayed data were corrected for diurnal geomagnetic field fluctuations, and we used the International Geomagnetic Reference Field to determine the magnetic background at the latitude of Kilbourne Hole. Further, a reduction-to-pole (RTP) operation was applied to recalculate the magnetic intensity values as if the inducing magnetic field had a 90° inclination [24].

The magnetic anomaly map is plotted in Figure 6. From this map, the following observations were made: (1) buried basaltic flows and dikes; (2) the Camp Rice formation and older units that predate the eruption; and (3) pyroclastic deposits associated with the Kilbourne Hole eruption. The measured values are reported in nT; the magnetic values that suggest an anomaly between point (1) and (2) fall in the range of ~600–800 nT, whereas between (1) and (3), the magnetic values are much lower (~100–300 nT).

Figure 6 graphically illustrates magnetic anomaly (a); the C_Rn_ region at the center and the magnetic anomaly in (b). The contours define an intersecting region with >85% overlap.

### 4.4. Data Analysis by Map Differences

To compare quantities of different units and scales, a normalization technique to a common range from 0 to 1 was conveniently applied. Missing values were obtained by interpolating experimental data. Figure 7a shows the difference heat map obtained by subtracting the magnetic values from the C_Rn_ values, and Figure 7b shows the gravity values subtracted from the C_Rn_ values.

The pixel maps in Figure 7a,b correspond to the central region of Kilbourne Hole, delimited by 313,500 m to 315,000 m in latitude and 3,538,000 m to 3,539,500 m in longitude; for convenience, the delimited areas are divided into twelve colored pixels or unit areas in each direction. Figure 7a clearly shows an overlap of small differences in the diagonal region from the lower left corner to the upper right corner. Figure 7b, on the contrary, shows much larger differences on the upper left corner and the surrounding regions.

To highlight the outcome resulting from this study, we report in Figure 8 a bar histogram related to the differences first between the C_Rn_ and magnetic anomaly overlap, and then the C_Rn_ and gravimetric data.

Figure 8 illustrates, in a compact form, the degree of correlation between the two sets of data. The first, i.e., the C_Rn_–magnetic data, demonstrate a variation between −0.5 and 0.25. This is in opposition to the second histogram, where the C_Rn_–gravity variations (−0.67 and +0.34) are larger. The overlapping histograms provide a means of identifying the degree of pair (Rad–Grav and Rad–Mag) correlation; a flat histogram that closes to a rectangular shape, corresponding to Rad–Grav, suggests a low correlation. The opposite is true for a Rad–Mag histogram with a higher observable value.

## 5. Conclusions

The measured radon activity of Kilbourne Hole demonstrated values in the interval 0.3–15.0 kBq/m^3^. We identified three well-separated radon anomalies associated with geological features (c.f. Figure 3). (1) At the center of the crater, probably related to the diatreme filling of breccia and dikes (volcanic intrusion); (2) at the western part of the crater, possibly associated to the soil’s natural radioactivity, to the presence of a fault, or both; and (3) at southern part of the crater, which is undoubtedly due to an inactive fault.

The Rn anomalies were contrasted to existing gravity and magnetic measurements. The high radon values were found to almost match the high magnetic anomaly region with an overlapping degree of >85%. This provided a positive correlation of the radon gas surficial activity concentration to the pyroclastic deposit. The maar formation could have occurred with a large mass rich in natural U, which could have produced the observed radioactive anomaly.

It is important to mention that a negative correlation between the NORM hot spots and the magnetic susceptibility of the rocks was reported in the past, in opposition to the results obtained in this study. In our case, this could be explained by the site’s different soil texture or the presence of a geological anomaly. It must be mentioned that comparing our findings to other studies is difficult, as the published literature on the subject is quite scant. Rusov et al. [8], for instance, observed that the magnetism diminished in the time interval of four years while the radon activity concentration increased.

Gravity values that were above-average for the site were found inside the crater where lower radioactivity surges were measured; this indicates a geological stratum of denser structure at the crater central section, de facto increasing the soil’s impedance to gas flow. This justifies the previously mentioned low correlation and, most of all, the dispersed “hot spots” specified in Figure 3. A dike formation beneath the center of the crater would support the assumptions of our findings and would explain the radon concentrations.

The radon anomaly located at the western part of the crater of Kilbourne Hole is attributed to an undisclosed geological fault, while the radon concentration at the south of the crater is correlated with a known inactive fault.

Our study points to the importance of studying the underground radon activity at volcanic craters such as the Kilbourne Hole. Future work will expand this work along the Potrillo volcanic field, which contains a number of young flows, three cinder cones, and two more maar volcanoes. We also are interested in furthering the correlation of a radon map with geomagnetic field and magnetic anisotropy to establish the soundness of the method.

## Figures and Tables

**Figure 1 ijerph-20-05185-f001:**
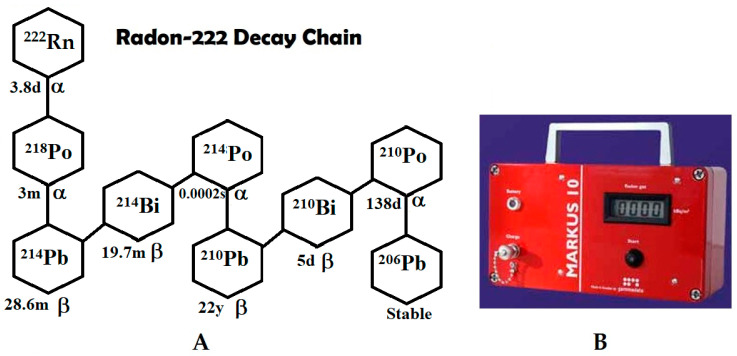
(**A**) Radon decay chain. Gaseous ^222^Rn decays with a half-life of 3.8 d into ^218^Po, which decays by alpha emission in three minutes [17]. (**B**) Gammadata’s Markus 10 instrument used to measure the radon concentration in soil.

**Figure 2 ijerph-20-05185-f002:**
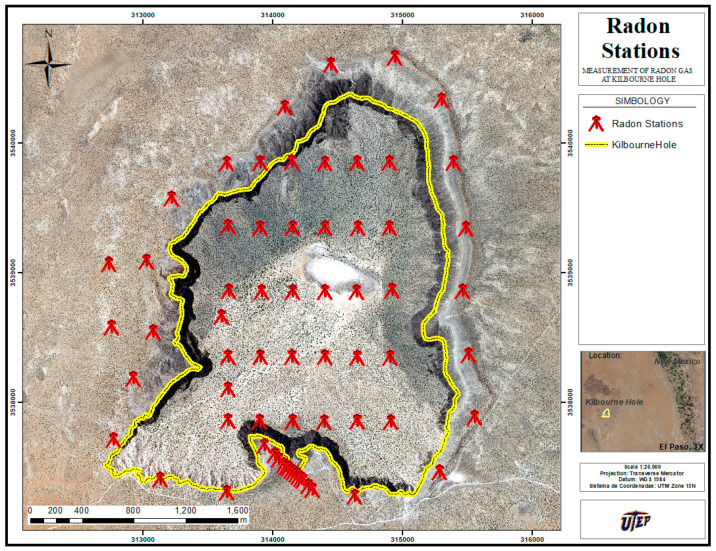
Map showing selected study site and soil Rn gas test points in red symbols; the region of interest is delimited by the yellow line circling the Kilbourne Hole sector.

**Figure 3 ijerph-20-05185-f003:**
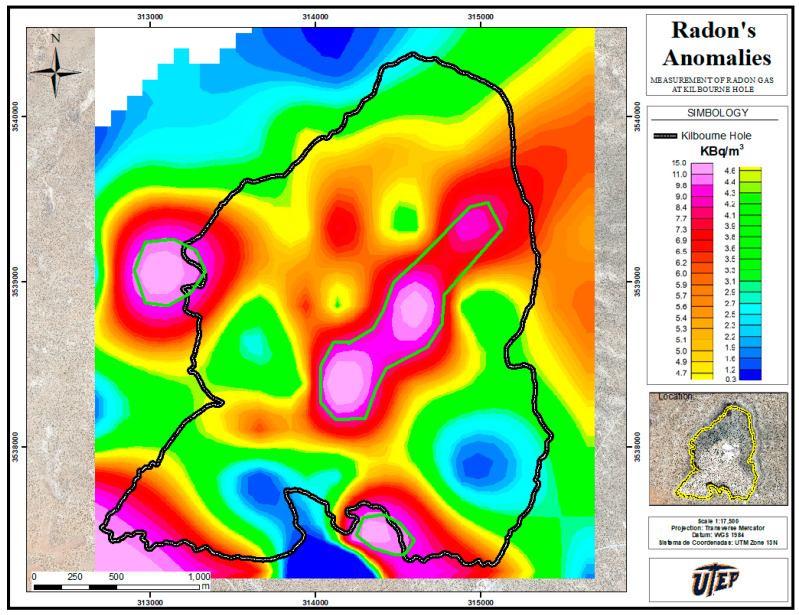
Heat map to evidence the C_Rn_ gradient; radon anomaly corresponds to darker colors, which vary from 0.3 kBq/m^3^ (blue) to 15.0 kBq/m^3^ (pink); the green lines represent the radon’s anomalies.

**Figure 4 ijerph-20-05185-f004:**
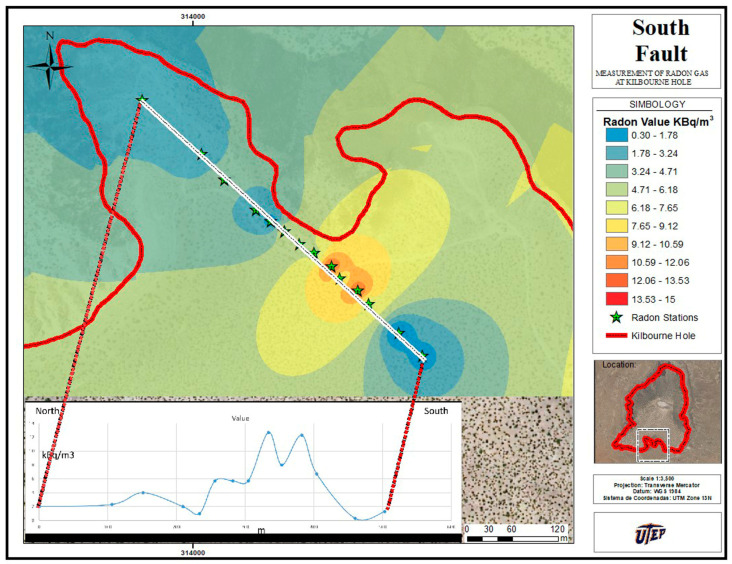
Soil Rn gas was measured along a transect across an inactive fault at Kilbourne Hole, south of the crater. To show the values departing from the average, measurements were taken at given space intervals. The corresponding curve is reported to provide insight into the anomaly.

**Figure 5 ijerph-20-05185-f005:**
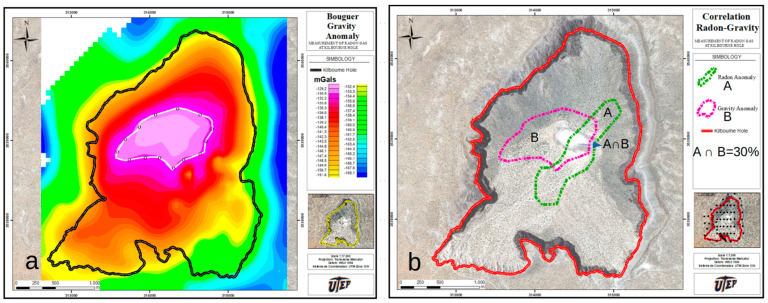
(**a**) Bouguer gravity anomaly map of Kilbourne Hole. The white, circled contour delimits the gravity anomaly, modified from the original data provided in [12]. (**b**) Anomaly of overlapping regions A∩B, related to Rn (A) and gravity data (B). It can also be seen that the overlapping extension of the crater surface (white central region), provided by the intersecting area (A∩B), covers half.

**Figure 6 ijerph-20-05185-f006:**
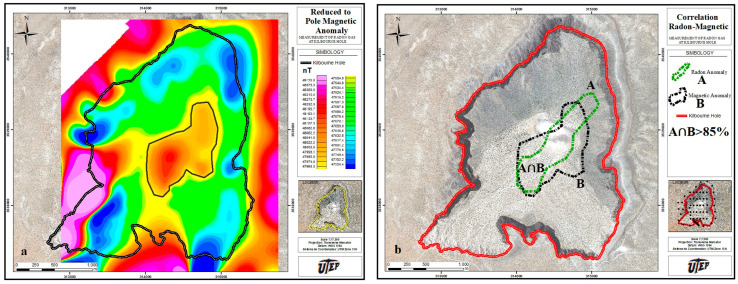
(**a**) RTP magnetic map of Kilbourne Hole, with black line delimiting the magnetic anomaly, from Maksim (2016). (**b**) C_Rn_ and magnetic field anomaly show the overlapping regions (>85%).

**Figure 7 ijerph-20-05185-f007:**
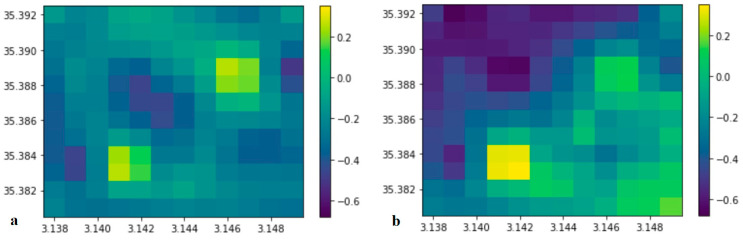
(**a**) Colored map to evidence differences between magnetic and the C_Rn_ values; (**b**) similarly for the gravity and the C_Rn_ values. The map abscissa is the latitude, and the ordinate is the longitude. Numbers near the column indicate the degree of similarity (lowest: 0.4; highest: −0.6).

**Figure 8 ijerph-20-05185-f008:**
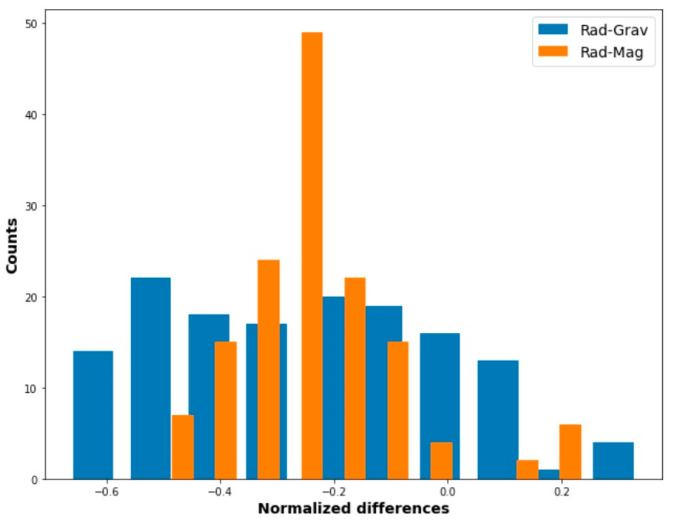
Bar histogram to highlight normalized differences between values of C_Rn_–gravity (with a tendency to a rectangular shaped distribution) and C_Rn_–magnetic data (showing heavier tails).

## Data Availability

Radon measurements are available by request to corresponding author.

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
