# Peer review of "Radon at Kilbourne Hole Maar and Magnetic and Gravimetric Correlations"

_ijerph, 2023, doi:10.3390/ijerph20065185_

Round 1

Reviewer 1 Report

In general, the main contents of the manuscript are correlation analysis between magnetic data and radon in-soil concentration and between gravimetric data and radon concentration in-soil at a specific site of volcanic field. Field measurement of radon concentration in-soil by grab sampling was carried out, results of radon concentration and its distribution is not given in detail, only shown by mapping, since the level of radon concentration itself is not the key point of the work.

This work could be concluded that radon in-soil can be applied as an effective tracer for geological researches. This work might be interesting to readers of geological science, but not very relevant to health.

Comments and suggestion:

1, Line 48: here 40Bq/m3 is the average level of indoor radon concentration worldwide, not comparable with radon concentration in soil.

2. Materials and Method: According to the contents of the first paragraphs of this section (line 79-111) , it is better to move them to the Introduction section.

3. Line 99: The level of outdoor radon concentration, 5-15 Bq/m3, nearly has no nothing to do with the level of radon in soil, it does not sound to mention here.

4. In Results section, Figure 2 is entitled “Soil Rn-gas anomaly map indicated by lower curve”. In the bottom part of the figure, how to see the higher curve? Might the title of the figure should be revised.

5. One observation of your work is “At the center the crater the highest radon values are reported”. Radon concentration is hardly influenced by water content of soil. So here, I wonder at the center of the crater, water content is also the highest or not, because of its rather lower terrain?

Author Response

Comments and suggestion:

1, Line 48: here 40Bq/m3 is the average level of indoor radon concentration worldwide, not comparable with radon concentration in soil.

The statement was modified, see line 53-57.

  1. Materials and Method: According to the contents of the first paragraphs of this section (line 79-111) , it is better to move them to the Introduction section.

Content is now in the Introduction.

  1. Line 99: The level of outdoor radon concentration, 5-15 Bq/m3, nearly has no nothing to do with the level of radon in soil, it does not sound to mention here.

Comment was deleted.

  1. In Results section, Figure 2 is entitled “Soil Rn-gas anomaly map indicated by lower curve”. In the bottom part of the figure, how to see the higher curve? Might the title of the figure should be revised.

The figure is now Fig. 3 and the caption of the figure was modified.

  1. One observation of your work is “At the center the crater the highest radon values are reported”. Radon concentration is hardly influenced by water content of soil. So here, I wonder at the center of the crater, water content is also the highest or not, because of its rather lower terrain?

A separate paragraph (the 3rd one) was added to Section 4.1 and two references were listed.

Reviewer 2 Report

Dear Authors,

The manuscript is well written and could address all concerns regarding the research topic and problems. 

1. I strongly advise to reduce the conclusion volume and remove the repeated sentences. Some part needs to move to discussion and some related info needs to be deleted.

2. it would be better to use middle dot between units instead of dot.

Overall, manuscript is in good condition and i would suggest a minor revision.

Regards,

Author Response

Comments and Suggestions for Authors

Dear Authors,

The manuscript is well written and could address all concerns regarding the research topic and problems.

  1. I strongly advise to reduce the conclusion volume and remove the repeated sentences. Some part needs to move to discussion and some related info needs to be deleted.

The Section “Conclusions” was modified exhaustively and shortened in size.

  1. it would be better to use middle dot between units instead of dot.

Problem was solve by using kBq/m3.

Overall, manuscript is in good condition and i would suggest a minor revision.

Regards,

Reviewer 3 Report

Title: A POSSIBLE CORRELATION OF RADON PRONE AREA 2 WITH MAGNETIC ANOMALY AT KILBOURNE HOLE MAAR, NEW MEXICO, USA

Comments:

Title not attractive. Please revise.

Abstract is very short and not informative. Please revise. Highlight the contribution and add some numerical results.

Add some recent reference to improve the literature part.

The research gap not defined.

Add some specific contributions at the end of introduction.

Add one figure to explain the methodology

The results part is very short

The part 4.1 Soil radon gas surficial survey need more explanations

The conclusion is very long and not focused

Please add the future work

Author Response

Comments and Suggestions for Authors

Title: A POSSIBLE CORRELATION OF RADON PRONE AREA WITH MAGNETIC ANOMALY AT KILBOURNE HOLE MAAR, NEW MEXICO, USA

Comments:

  • Title not attractive. Please revise.

Title was shortened to “RADON AT KILBOURNE HOLE MAAR AND MAGNETIC AND GRAVIMETRIC CORRELATIONS”

  • Abstract is very short and not informative. Please revise. Highlight the contribution and add some numerical results.

The Abstract was modified according to the instructions of the reviewer.

  • Add some recent reference to improve the literature part.

We added refs. 3 and 19.

  • The research gap not defined.

The research gap was added as a new paragraph at the end of Section 1.

  • Add some specific contributions at the end of introduction.

A list of novelties was added at the end of the Introduction just before Section 1.1.

  • Add one figure to explain the methodology

A new figure 1 was used to further explain the methodology used in the measurement of the underground Rn.

  • The results part is very short

Old Figure 1 was divided into new figures 2 and 3, and their description was expanded in the Results section.

  • The part 4.1 Soil radon gas surficial survey need more explanations

The explanations of Section 4.1 were expanded adding two more paragraphs.

  • The conclusion is very long and not focused

The Conclusion was rewritten in a more succinct and focused manner

  • Please add the future work

A paragraph was added at the end of Section 5.

Reviewer 4 Report

The review topic is really interesting and the manuscript is well-written. Therefore, the manuscript has some problems that are listed below:

1) What are the novelties of your study? Please, add them at the end of the 1.1 section. Is it the first time that the correlation between radon gas concentration and local geophysical characteristics are reported in similar sites?

2) How long were the measurements (e.g., Crn) collected? Please, add in the material and methods section.

3) Please, add the correlation and statistical analysis used in section 4.4 in the Material and methods. Which type of correlation was used?

4) Please, compare your results with previous studies. Did the observed correlation find also in other studies?

Author Response

Comments and Suggestions for Authors

The review topic is really interesting and the manuscript is well-written. Therefore, the manuscript has some problems that are listed below:

1) What are the novelties of your study? Please, add them at the end of the 1.1 section. Is it the first time that the correlation between radon gas concentration and local geophysical characteristics are reported in similar sites?

A list of novelties was added at the end of the Introduction just before Section 1.1. A comment and a reference about previous similar studies was added in Section 5 (see line 312).

2) How long were the measurements (e.g., Crn) collected? Please, add in the material and methods section.

The times are listed in Section 2, point iii.

3) Please, add the correlation and statistical analysis used in section 4.4 in the Material and methods. Which type of correlation was used?

Unfortunately, Pearson (the usual correlation tool) does not work with two-dimensional data. Our search for correlation is simply a physical geography, i.e. an overlap of the maps to determine the percentage of the crater area with high radon concentration that overlaps with high magnetic anomaly (85%), and with high gravimetric data (30%). We added a comment about this at the end of Section 4.1.

4) Please, compare your results with previous studies. Did the observed correlation find also in other studies?

A comment and a reference were added in the Conclusions (see line 312).

Round 2

Reviewer 3 Report

The paper can be accepted 

Author Response

No new points were raised. Thanks for accepting the last version.